# Behavior Modeling Space Reconstruction for E-Commerce Search

## ABSTRACT

Delivering superior search services is crucial for enhancing customer experience and driving revenue growth in e-commerce. Conventionally, search systems model user behaviors by combining user preference and query-item relevance statically, often through a fixed logical 'and' relationship. This paper reexamines existing approaches through a unified lens using both causal graphs and Venn diagrams, uncovering two prevalent yet significant issues: entangled preference and relevance effects, and a collapsed modeling space. To surmount these challenges, our research introduces a novel framework, DRP, which enhances search accuracy through two components to reconstruct the behavior modeling space. Specifically, we implement preference editing to proactively remove the relevance effect from preference predictions, yielding untainted user preferences. Additionally, we employ adaptive fusion, which dynamically adjusts fusion criteria to align with the varying patterns of relevance and preference, facilitating more nuanced and tailored behavior predictions within the reconstructed modeling space. Empirical validation on two public datasets and a proprietary e-commerce search dataset underscores the superiority of our proposed methodology, demonstrating marked improvements in performance over existing approaches. The code is available at https://anonymous.4open.science/r/DRP-ABE1/.

**Relevance Statement**: This paper studies the reconstruction of collapsed behavior modeling space with both preference editing and adaptive fusion, ultimately enhancing e-commerce search. Thus, this paper can be relevant to topics 'Web search models and ranking', 'Ad search and search for Web retail', and 'Personalized Search'.

## KEYWORDS

Behavior Modeling; E-Commerce Search

**ACM Reference Format:**
Anonymous Author(s). 2025. Behavior Modeling Space Reconstruction for E-Commerce Search. In *Proceedings of Proceedings of the ACM Web Conference 2025 (WWW'25)*. ACM, New York, NY, USA, 9 pages. https://doi.org/10.1145/nnnnnnn.nnnnnnn

## 1 INTRODUCTION

E-commerce has revolutionized the retail industry by harnessing the capabilities of the World Wide Web to offer consumers round-the-clock access to a global array of products and services [19]. In e-commerce applications, the search functionality serves as a critical determinant, matching and ranking relevant items in response to user queries, thereby driving user engagement (e.g., clicks). Consequently, a fundamental objective of search model development is to accurately and comprehensively understand user behaviors (i.e., why users interact with certain presented items instead of others). However, in practice, available data often only indicates whether a user interacted with an item, without providing explicit insights into the underlying motivations. Thus, effectively modeling user behaviors remains one of the most significant technical challenges in e-commerce search optimization.

User behaviors in e-commerce search can be majorly attributed to the query-item relevance and the user preference on items [7] since the behavior data usually consists of candidate items, users, and their queries for ongoing search sessions. Accordingly, previous studies can be divided into three research lines: relevance-only modeling, preference-only modeling, and the joint of both. A fundamental assumption of the first two approaches is that utilizing relevance or preferences provides sufficient information to model user behaviors and explains why users interact with particular items. Relevance-only methods regard e-commerce search as a basic user-independent search problem and focus on capturing the relevance between queries and items. As such, established ad-hoc search methodologies, such as BM25 [26] and DSSM [18], can be effectively applied as the relevance-only strategy. Some researchers have suggested that relevance effects on user behaviors are highly personalized and advanced relevance-only methods by integrating user features for personalized search systems [4, 5, 10]. While these methods adopt a search perspective that prioritizes relevance, an alternative viewpoint considers the query as an integral part of the user features, reflecting user intent for a particular search session. Under this assumption, recommendation models like DFM [14] and Wide & Deep [9], which better capture user preferences for items, are usually deployed as preference-only methods. The first two lines tend to model relevance or preference effects individually and are trained with behavior signals, suffering from the misaligned training [24]. We will further illustrate this mismatch as the concept of collapsed modeling space in Section 2.2 based on a Venn diagram.

Additionally, these models typically produce a single relevance or preference prediction, while neglecting either effect could lead to suboptimal outcomes, as highlighted by [7]. For example, prioritizing preference alone can create an echo chamber [11], where favored yet irrelevant items are ranked higher than more relevant ones. To address this, an instant solution is to include both effects for joint modeling. Most existing joint frameworks focus on developing pre-trained relevance models [8, 20, 33–35]. In these frameworks, the relevance model is initially trained on human-labeled datasets that consist of query-item relevance signals. Subsequently, they are integrated with the preference model for joint optimization by the behavior data and finally deployed in search systems. However, this approach still faces two challenges: **i) Biased and collapsed modeling space.** Since the relevance model is trained before the preference model, the combined results may disproportionately

favor relevant items, resulting in a biased modeling space. In addition, the current scoring formula is static, lacking the flexibility to adapt to the diverse patterns of samples with different relevance-preference statuses, finally collapsing the model space. We will illustrate this point with a Venn diagram in Section 2.2. **ii) Laborious data collection.** Pre-trained relevance models heavily depend on exhaustively collected human-labeled relevance data, making them difficult to apply universally across various e-commerce platforms, particularly for smaller companies that cannot afford such labor-intensive data collection. *Thus, there is a pressing need for a joint modeling framework that effectively captures user behavior within the correct modeling spaces, while also possessing adaptive scoring capabilities without relying on human-labeled data.*

Furthermore, existing methods overlook the inherent influence of relevance on user preference as visualized by the causal graph in Figure 1 (a)[1]. This unaddressed influence can lead to impure preference prediction, further collapsing the modeling space and finally impairing the joint behavior modeling. To tackle these, this paper introduces **DRP**, which models user behavior through **D**isentangling **R**elevance and **P**reference effects, alongside an adaptive fusion formula. Specifically, DRP begins with eliminating the relevance effect from preference predictions by editing corresponding representations, thus reconstructing the fine-grained modeling space. DRP subsequently learns nuanced user behaviors, such as item clicks, based on the dual-level adaptive integration within the reconstructed modeling space. Our major contributions are as follows:

- We build a theoretical framework based on causal graphs and Venn diagrams to comprehensively review existing behavior modeling approaches for e-commerce search.
- We explicitly divide the behavior modeling space into six areas, based on which we point out the collapsed modeling space caused by three problems of existing methods.
- We introduce DRP, effectively addressing the collapsed modeling space with disentangled relevance and preference effects, as well as dual-level adaptive fusion. Notably, DRP operates without the need for human-labeled relevance data, making it deployable with just behavior data.
- We perform extensive experiments on two large public datasets and one private e-commerce search dataset, demonstrating the superior performance of DRP.

## 2 PRELIMINARY

In this section, we begin by introducing key notations and joint modeling structures. Following this, we will review existing methods and highlight limitations with a causal graph and a Venn diagram.

### 2.1 Joint Modeling Framework

In this section, we outline the notation and fundamental structure of the joint modeling framework employed in this paper.

The structure is visually represented in Figure 2(a). Typically, the textual information from the input query is transformed into dense representations using a text encoder. These semantic representations, along with additional query features such as query frequency, are then synthesized into the query embedding $q$. Likewise, the text

---

[1]More details will be provided in Section 2.2.

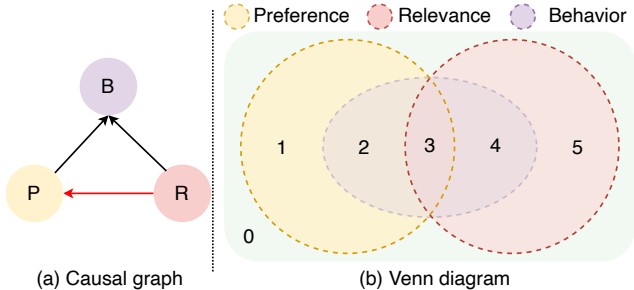

**Figure 1: Causal graph and Venn diagram for behavior modeling in e-commerce search.**

**Table 1: Area indicators for Venn diagram.**

| Area# | 0 | 1 | 2 | 3 | 4 | 5 |
|-------|---|---|---|---|---|---|
| P | 0 | 1 | 1 | 1 | 0 | 0 |
| R | 0 | 0 | 0 | 1 | 1 | 1 |
| B | 0 | 0 | 1 | 1 | 1 | 0 |

associated with items—such as titles or product names—is also converted into dense representations with the same text encoder and then combined with other item features to form the item embedding $v$. For user data, we integrate users' static features as 'Activity Degree' and encoded behavioral sequences features as 'Clicked Items' into the user embedding $u$.

Then, the relevance prediction $\hat{r}$ is generated through the relevance model, denoted as RM. For ad-hoc search methods [18, 26], it can be mathematically expressed as:

$$\hat{r} = \text{RM}(q, v) \tag{1}$$

In the case of personalized search methods [1, 6], the relevance model incorporates the user embedding to facilitate personalization, yielding $\hat{r} = \text{RM}_u(q, v)$.

The user preference for a specific item under the query can likewise be predicted using the preference model PM:

$$\hat{p} = \text{PM}(q, v, u) \tag{2}$$

The query embedding $q$ is consistently provided as input to the preference model, as it encapsulates the user's intent for the current session—an essential aspect of preference modeling.

Finally, the relevance and preference predictions are integrated to produce the final score, typically calculated via multiplication::

$$\hat{y} = \hat{r}^{\delta} \cdot \hat{p} \tag{3}$$

where $\hat{y}$ represents the final behavior prediction, $\delta$ is a hyper-parameter that adjusts the balance between relevance and preference. For relevance-only modeling, we assume $\hat{p} \equiv 1$, while for preference-only modeling, $\hat{r} \equiv 1$.

### 2.2 Collapsed Modeling Space Problem

This section will first introduce the causal graph and Venn diagram for behavior modeling in e-commerce search. Then we point out the collapsed modeling space of existing methods, and finally illustrate our motivations in detail.

The causal graph is illustrated in Figure 1(a), where $B$ represents user behavior—such as clicks and purchases—while $P$ and $R$ denote

preference and relevance, respectively. In this graph, the black arrows $P \rightarrow B$ and $R \rightarrow B$ indicate the preference and relevance effect on user behaviors. It is clear that users would respond to the returned items based on these effects: they may click on items with high relevance, make purchases due to strong preferences, or express dissatisfaction when faced with low relevance. Besides, the preference is also influenced by relevance (the red arrow $R \rightarrow P$). The reason is that the preference is inherently subjective. When the item is highly relevant to the user query, the preference score may increase, reflecting a stronger affinity for the item in the context of the specific query—even if the user typically does not favor it. **For instance**, a user who prefers a formal dressing style may buy 'Nike Shoes' because they are highly relevant when searching for 'Sports Shoes'. In this case, the model might learn the user's preference for 'Nike Shoes' or the Nike brand although the user generally prefers formal dressing brands. This inflated preference effect can finally harm the search system. In the same example, the next time this user looks for a dress for everyday outfits, where sports styles aren't favored, the model might still suggest 'Nike Dress' because it is relevant to the search and is consistent with the biased preference on 'Nike' learned from the earlier session.

Existing joint modeling frameworks primarily concentrate on the effects of relevance and preference on user behaviors ($P \rightarrow B$ and $R \rightarrow B$), overlooking the influence of relevance on preference signals ($R \rightarrow P$). Based on the assumption that only $P \rightarrow B$ and $R \rightarrow B$ exist in Figure 1(a), we visualize the behavior modeling space with a Venn diagram in Figure 1(b). In this diagram, the green rectangle encompasses all behavior data, the yellow circle represents user preferences, the red circle denotes positive relevance effects, and the purple ellipse highlights positive behaviors. We label six distinct areas with numbers 0-5, with corresponding preference, relevance, and behavior signals detailed in Table 1. For example, in click modeling, Area #0 corresponds to the situation where $P = 0, R = 0, B = 0$. This indicates that the item is neither relevant to the query nor favored by users, and with a negative click signal. We divide the Venn diagram into six areas as $P = 0, R = 0, B = 1$ and $P = 1, R = 1, B = 0$, contradicting our causal assumption that the behavior is only attributed to the relevance and preference.

We suggest that current works typically suffer from the collapsed modeling space. For relevance-only or preference-only methods [3, 31], the single relevance and preference effects are trained with the behavior signals, leading to the modeling space mismatch. Specifically, the relevance model aims to learn the relevance effect (Area#3-5), while its training signals are derived from the behavior space (Area#2-4). Consequently, the relevance model may misclassify Area#2 as relevant and Area#5 as irrelevant, contrary to the actual situation. This similar misalignment also occurs in preference-only methods, where Area#4 is incorrectly perceived as a preference and Area#1 as the dislike.

Besides, there are three problems for joint models based on the modeling space. **Problem 1:** The neglect of the relevance influence on the preference of joint modeling methods can collapse the modeling space, as it may ruin the preference modeling [15]. In the worst case, we can imagine that only relevant items can be preferred by users, where Area#1&2 vanish. **Problem 2:** Existing joint models [16, 35] usually integrate two effects with a fixed formula as in Equation (3). However, this static integration falls short of

capturing the complexities of modeling space, which can exhibit diverse relevance-preference patterns. Equation (3) only learns the positive behavior with positive preference and relevance effects. As a result, it would only distinguish Area#0&3, missing the pattern of Area#2&4 and Area#1&5. **Problem 3:** Current optimization scheme that trains the relevance model before integrating it with the preference model [8, 35] may also collapse the modeling space. Under this pipeline, the joint model first identifies the relevant space (Area#3-5). During the subsequent training of preference effect, the model may prioritize relevant items over preferred items in any situation. For instance, comparing Area#4 and Area#2 reveals that Area#4 can be learned more easily due to the pre-trained relevance knowledge, while the model may miss Area#2 because it contradicts this prior knowledge, e.g., irrelevant but clicked. Additionally, when we compare Area#2 and Area#3, it becomes evident that prior relevance knowledge can disrupt preference model training. Both Area#2 and Area#3 exhibit the same preference and behavioral signals ($P = 1, B = 1$), yet opposing relevance.

To overcome these problems, we propose a preference editing approach designed to eliminate the relevance influence on the preference (the red arrow in Figure 1(a)), grounding the basis for reconstructing the desired modeling space as in Figure 1(b) to solve Problem 1. Furthermore, we introduce an adaptive fusion method that enables the joint model to effectively learn and capture nuanced patterns within the reconstructed modeling space to overcome Problem 2. With the proper modeling space and fine-grained learning ability, the proposed DRP is able to learn independent relevance and preference effects with only behavior data in an end-to-end manner, solving Problem 3.

## 3 METHOD

We introduce technical details of the proposed DRP in this section. We begin with an overview, followed by a detailed illustration of key components, as well as the processes of inference and optimization.

### 3.1 Overview

This section briefly introduces the overall process of DRP with the framework overview illustrated in Figure 2.

Figure 2 (a) presents the basic structure of the joint modeling frameworks introduced in Section 2.1. The joint model first obtains the query and item representations, $q$ and $r$, using the text encoder and feature encoder, along with the user representation $u$ from the sequence encoder and feature encoder. Subsequently, relevance and preference effects are predicted based on these representations, as described in Equation (1) and Equation (2). These predictions are then merged to generate the behavior prediction, as shown in Equation (3). The proposed DRP operates on the last layer representation of the preference model (within the orange dashed block) and the scoring formula for behavior prediction (within the green dashed block), without constraining the underlying feature encoders or backbone models. Specifically, we design a preference editing framework to eliminate the relevance influence from the preference prediction, as illustrated in Figure 2 (b). We use the last layer representations ($e_p, e_r$) as neural abstractions for corresponding causal notions ($P, R$), and identify an orthogonal low-rank matrix $O$ that best matches the influence space of $R \rightarrow P$. By editing

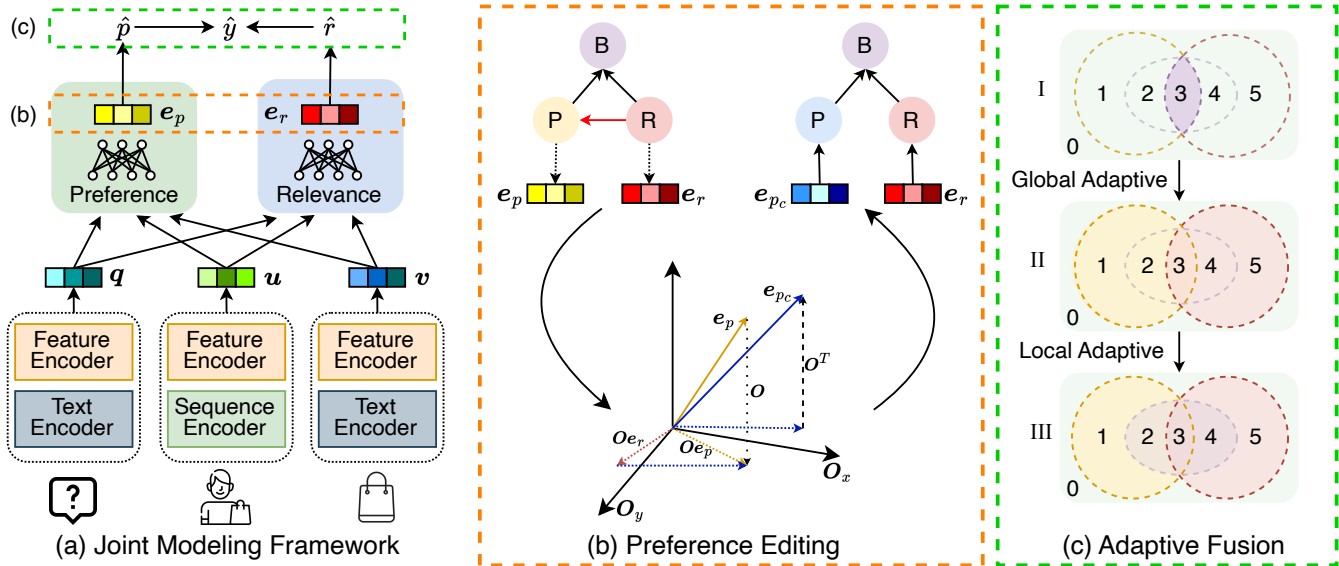

Figure 2: Framework Overview. The joint modeling framework is depicted in (a). The preference editing is represented in (b). Adaptive fusion is illustrated in (c).

the preference representation in this space and converting the result back to the original space using $O^T$, we can effectively predict user behavior, capturing both direct relevance and preference effects, as shown by the modified causal graph transitioning from the top left to the top right in the figure. The disentanglement of relevance and preference further allows us to reconstruct a fine-grained modeling space, as depicted in Figure 2 (c). Here, we correctly segment the modeling space into six areas and separately learn their distinct patterns using dual-level adaptive fusion.

## 3.2 Preference Editing

In this section, we first mathematically formulate the ideal behavior modeling method based on relevance and preference effects, then introduce a preference editing method to eliminate the relevance influence on effect to achieve this.

Existing joint modeling frameworks typically combine the estimated effects $\hat{p}, \hat{r}$ as in Equation (3). These frameworks assume that there are only direct effects of $P, R$ on $B$, i.e., only black arrows exist in the causal graph (Figure 1 (a)). However, they overlook the relevance effect on preference (the red arrow), which can produce the indirect relevance effect on user behaviors (the path $R \rightarrow P \rightarrow B$ in the graph), potentially leading to biased outcomes, as highlighted by Guo et al. [15]. Therefore, the ideal modeling formula should be:

$$\hat{p}_c = f(\hat{p}) - g(\hat{r}) \tag{4}$$

$$\hat{y} = \hat{r}^\delta \cdot \hat{p}_c \tag{5}$$

where $\hat{p}_c$ represents the calibrated preference prediction, which mitigates the relevance effect in the initial prediction $\hat{p}$ as in Equation (4), effectively removing the indirect relevance effect on behaviors. In Equation (4), $f$ and $g$ are specifically designed functions to achieve the disentanglement purpose, culminating in a pure preference prediction in Equation (5).

To find the optimal functions $f$ and $g$, which operate on the causal predictions $\hat{p}$ and $\hat{r}$, we can align these high-level causal signals with the low-level neural representations in deep models. This alignment allows us to alternatively determine the optimal transformations on neural representations in a learning-based manner [12]. Specifically, we treat the hidden representation from the final layer of preference models as the base representation and that from relevance models as the source representation. According to Geiger et al. [13], a low-rank neural space can be identified that best matches the causal intervention space from the source effect to the base. Importantly, modifying the base representation within this low-rank space preserves information unrelated to the intervention. Building on this insight, we aim to learn the optimal orthogonal low-rank projection matrix as the desired transformation:

$$e_{p_c} = O^T(Oe_p - Oe_r) \tag{6}$$

Let $e_{p_c}, e_p,$ and $e_r$ represent the final layer's hidden representations for $\hat{p}_c, \hat{p},$ and $\hat{r}$, respectively. We introduce $O \in \mathbb{R}^{H \times D}$, a learnable low-rank projection matrix with orthogonal rows, where $H$ is the dimension of the last layer, and $D$ is the dimensionality of the low-rank subspace. Since $O$ is orthogonal, $O^T$ serves as its inverse, allowing the conversion of the edited representation back into the original space. The calibrated preference can then be computed based on the edited preference representation:

$$\hat{p}_c = W_p e_{p_c} + b_p \tag{7}$$

where $W_p$ and $b_p$ are the transformation matrix and bias for the linear decoder for preference prediction, respectively.

The process of preference editing is visualized in Figure 2 (b). We begin by setting $e_p$ and $e_r$ as low-level representations for the high-level causal notions $P$ and $R$, respectively. Next, we learn an orthogonal space $O$ that best aligns with the intervention space of $R \rightarrow P$. In the figure, $O \in \mathbb{R}^{3 \times 2}$ is illustrated as an orthogonal projection matrix, with the corresponding space represented as

a two-dimensional surface spanned by $O_x$ and $O_y$. Within this intervention space, we subtract the relevance influence ($Oe_r$, shown as the red dotted vector) from the preference representation ($Oe_p$, the yellow dotted vector), obtaining the unbiased user preference (the blue dotted vector). We then project it back to the original behavior modeling space using $O^T$, allowing us to decode it for unbiased preference prediction. This process culminates in the top-right causal graph, which directly models both preference and relevance based on behavioral signals and allows the segmentation of the modeling space as Figure 1 (b).

## 3.3 Adaptive Fusion

This section introduces a dual-level adaptive fusion method to guide the joint model toward learning from the appropriate modeling space, addressing Problem 2 discussed in Section 2.2.

From Equation (3) and Equation (5), we observe that existing scoring methods will predict positive user behavior only when $\hat{p}_c(\hat{p}) \to 1$ and $\hat{r} \to 1$. Equation (5) can be rewritten as:

$$
\begin{aligned}
\hat{y} = \hat{r}^\delta \cdot \hat{p}_c &= \hat{r}^{\delta-1} \cdot \hat{p}_c \hat{r} \\
&= \hat{r}^{\delta-1} \cdot \begin{pmatrix} 1 & 0 \end{pmatrix} \begin{pmatrix} \hat{p}_c \\ 1 - \hat{p}_c \end{pmatrix} \begin{pmatrix} \hat{r} & 1-\hat{r} \end{pmatrix} \begin{pmatrix} 1 \\ 0 \end{pmatrix} \\
&= \hat{r}^{\delta-1} \cdot \begin{pmatrix} 1 & 0 \end{pmatrix} \begin{pmatrix} \hat{p}_c \cdot \hat{r} & \hat{p}_c \cdot (1-\hat{r}) \\ (1-\hat{p}_c) \cdot \hat{r} & (1-\hat{p}_c) \cdot (1-\hat{r}) \end{pmatrix} \begin{pmatrix} 1 \\ 0 \end{pmatrix}
\end{aligned}
\tag{8}
$$

Denoting the $2 \times 2$ matrix in Equation (8) as $\begin{pmatrix} \mathbb{P}_{11} & \mathbb{P}_{10} \\ \mathbb{P}_{01} & \mathbb{P}_{00} \end{pmatrix}$, we can find that $\mathbb{P}_{ij}$ is the prediction for the relevance-preference status $P = i, R = j$. For example, $\mathbb{P}_{10} = \hat{p}_c \cdot (1 - \hat{r})$ is the probability that this sample belongs to Area#1 or Area#2 where $P = 1, R = 0$ as listed in Table 1. Notice that only $\mathbb{P}_{11}$ will influence the final prediction, as the probability for the remaining areas is rendered as zero by left multiplication with $\begin{pmatrix} 1 & 0 \end{pmatrix}$ and right multiplication with $\begin{pmatrix} 1 \\ 0 \end{pmatrix}$.

The corresponding modeling space is visualized in Figure 2 (c-I), where only Area#3 ($\mathbb{P}_{11}$) is properly learned. In this Venn diagram, Area#1, 2 ($\mathbb{P}_{10}$) and Area#4, 5 ($\mathbb{P}_{01}$) are hard to distinguish as they are merged with Area#0 ($\mathbb{P}_{00}$). To model the nuanced pattern of these areas, the global adaptive fusion replaces $\begin{pmatrix} 1 & 0 \end{pmatrix}$ and $\begin{pmatrix} 1 \\ 0 \end{pmatrix}$ in Equation (8) with learnable parameters:

$$
\begin{aligned}
\hat{y}_g &= \hat{r}^{\delta-1} \cdot \boldsymbol{\alpha} \begin{pmatrix} \mathbb{P}_{11} & \mathbb{P}_{10} \\ \mathbb{P}_{01} & \mathbb{P}_{00} \end{pmatrix} \boldsymbol{\beta}^T \\
&= \hat{r}^{\delta-1} (\mathbb{P}_{11}\alpha_1\beta_1 + \mathbb{P}_{10}\alpha_1\beta_0 + \mathbb{P}_{01}\alpha_0\beta_1 + \mathbb{P}_{00}\alpha_0\beta_0)
\end{aligned}
\tag{9}
$$

where $\boldsymbol{\alpha} = (\alpha_1, \alpha_0)$ and $\boldsymbol{\beta} = (\beta_1, \beta_0)$ are learnable parameters, and $\hat{y}_g$ stands for the behavior prediction with global adaptive fusion. Since $\sum_{ij} \mathbb{P}_{ij} = 1$ and $\boldsymbol{\alpha}, \boldsymbol{\beta}$ are usually non-zero, Equation (9) can distinguish the differences of different patterns when there is a clear result. For example, suppose $\mathbb{P}_{01} = 0.99$, the joint model is able to independently learn the distinct pattern for Area#4 and Area#5 (the area for $P = 0, R = 1$).

However, the global adaptive fusion is still insufficient to model the entire diagram. It divides the modeling space into four parts based on the relevance-preference status, but users exhibit varying sensitivities to relevance, depending on different queries and items.

**Table 2: Dataset statistics.**

|          | # User | # Item    | # Query | # Interaction | # Session |
|----------|--------|-----------|---------|---------------|-----------|
| KuaiSAR  | 19,851 | 1,974,165 | 126,027 | 3,038,362     | 186,268   |
| JDSearch | -      | 8,305,606 | 110,763 | 15,439,583    | 173,825   |
| Private  | 9,426  | 196,645   | 123,941 | 45,245,013    | 1,165,596 |

This results in different behavior signals for the same relevance-preference status [16]. As illustrated in Figure 2 (c-II), the global adaptive fusion fails to distinguish between Area#1 and Area#2, as well as between Area#4 and Area#5. To address this issue, we further propose the local adaptive fusion:

$$
\hat{y}_l = \hat{y}_g \cdot \mathcal{F}(\boldsymbol{u}, \boldsymbol{v}, \boldsymbol{q}; \hat{y}_g, y)
\tag{10}
$$

where $\hat{y}_g$ is the global adaptive prediction from Equation (9), and $y$ is the ground truth of user behaviors (e.g., $y = 1$ denotes 'Click' for click modeling). $\mathcal{F}$ aims to fit the gap between the global score $\hat{y}_g$ and the heterogeneous behavior signals $y$ of similar relevance-preference statuses, ultimately achieving nuanced modeling as illustrated in Figure 2 (c-III). Note that only $\boldsymbol{u}, \boldsymbol{v}, \boldsymbol{q}$ are fed to $\mathcal{F}$ during the feed-forward loop, while $\hat{y}_g$ and $y$ are used to construct the training objective for $\mathcal{F}$ in the training loop. This fills the gap between the diverse $y$ and the similar global predictions $\hat{y}_g$ in Area#1&2 and Area#4&5, enabling end-to-end training and serving.

## 3.4 Inference & Optimization

We illustrate the inference and optimization process of DRP in this section. Overall, DRP operates on the scoring stage, where the preference model and the relevance model remain unaffected.

For inference, the relevance is the same as calculated in Equation (1). We replace the last layer representation of the preference model as in Equation (6) and predict the preference effect following Equation (7). Finally, we propose the dual-level adaptive fusion, merging two effects as in Equation (10).

With the disentangled relevance and preference effects and the well-segmented modeling space, DRP is able to learn fined-grained patterns without the human-labeled relevance signals in an end-to-end manner. As there are no auxiliary losses for DRP, we can optimize the framework with the mere behavior modeling loss and apply the gradient-descent strategy. Take the click modeling task (click-through rate prediction) as an example, we can train DRP with binary-cross entropy loss:

$$
\mathcal{L} = \sum_{y \in \mathcal{T}} y \cdot \log \hat{y} + (1 - y) \cdot \log (1 - \hat{y})
\tag{11}
$$

where we use $\mathcal{T}$ to denote the training set.

## 4 EXPERIMENT

### 4.1 Experiment Setting

We provide the experimental settings in this section, including details about datasets, relevance and preference backbones, baselines we compared, evaluation metrics, and implementation details.

*4.1.1 Dataset.* DRP is evaluated on two large-scale public datasets and data collected on our private e-commerce platform. **KuaiSAR** [28], is a unified search and recommendation dataset released by Kuaishou, a short-video platform. We split out the video

search log to test the effectiveness of DRP in scenarios other than e-commerce. **JDSearch** [23], constructed by JD.com, a popular online shopping platform. This is a typical e-commerce scenario. **Private**, collected from the one-month daily log on our private e-commerce platform. To construct a viable subset for offline evaluation and filter out long-tail users and items, we apply the core-100 algorithm proposed in Amazon Review [25]. We will publicize the dataset after the anonymization in the future. It is notable that all three datasets contain *real* queries (anonymized for public two). The detailed statistics, including the number of users, items, queries, interactions, and search sessions, are listed in Table 2. For data splitting, we split the sessions in the first eight days as the training set, the middle one day as the validation, and the last day as the test set for KuaiSAR. Similarly, search sessions in the first 80% time period are the training set, the middle 10% is the validation, and the remaining 10% is the test set for Private. JDSearch does not contain timestamps and is randomly split with the same ratio (8:1:1).

*4.1.2 Backbones & Baselines.* To verify the flexibility of our model, we integrate various relevance and preference backbone models for comparison. **Relevance Models**: DSSM [18] (ad-hoc), QEM [3] (ad-hoc) and HEM [3] (personalized). **Preference Models**: the most popular MLP [36], frequently deployed cross network DCN [30]. Besides, we compare several **Joint Modeling Baselines**: CLK [34], which suggests the relevance model can learn relevance from the click data. Compared with the joint modeling framework, this method adds an auxiliary loss to refine the training of relevance models based on the relevance signal constructed from the click data. NISE [17] and DCMT [37], which tackle modeling space problems for the CVR prediction. PRINT [16] includes a Personalized Relevance Incentive network to adaptively fuse the relevance and preference predictions for different samples.

*4.1.3 Evaluation Metrics.* Following previous works [14], we adopt Area Under the ROC Curve, denoted as **AUC** ($\uparrow$), and **LogLoss** ($\downarrow$) as the evaluation metrics. These two metrics take all positive and negative samples into account to reveal the discriminative capabilities of the models. Besides, to show the performance in ranking, we also consider Top-10 Hit Rate, *i.e.,* **HR** ($\uparrow$) and Top-10 Normalized Discounted Cumulative Gain, *i.e.,* **NDCG** ($\uparrow$) as metrics. It is worth noting that we only take the sessions with positive samples into account when calculating these two metrics.

*4.1.4 Implementation Details.* For the basic joint modeling structure, we employ M3E [32] to encode textual contents in Private. We encode the anonymous text within JDSearch and KuaiSAR, as well as user behavior sequences in all datasets following UniSAR [27]. Other features are all encoded as one-hot signals and converted to embeddings. Above this, we obtain 64-dimensional $q, v, u$ and feed to the relevance and preference model. Units of the prediction layer, i.e., the last MLP component of RM, PM, is [64, 32, 1], indicating $H = 32$ for $\boldsymbol{e}_p, \boldsymbol{e}_{p_c}, \boldsymbol{e}_r$. The low-rank projection space is 16-dimensional, i.e., $D = 16$ for $\boldsymbol{R}$. For global adaptive fusion, the initial state is $(1, 0.5)$ for $\boldsymbol{\alpha}$ and $\boldsymbol{\beta}$.

## 4.2 Overall Performance

This section comprehensively assesses DRP on various relevance and preference backbones, comparing it with existing methods crafted to enhance the joint modeling. Each experiment is repeated ten times with different seeds for reliable results. We present the

mean of these ten trials with four valid decimal places, and implement the one-tailed t-test to confirm the superiority of **the best-performing method** (in bold) over the runner-up (underlined). The result is presented in Table 3, where 'PM' denotes the preference backbone, 'RM' is the relevance model, 'JM' stands for the method to boost joint modeling frameworks, and '-' means no solution is applied. For example, the first three rows that with 'PM' and 'JM' as '-' are relevance-only methods.

Overall, the joint modeling approach outperforms methods that consider only relevance or preference due to its integration of both factors. This superiority underscores the importance of developing joint modeling frameworks. Additionally, we observe that the personalized relevance model and the preference model exhibit similar performance levels (for example, MLP versus HEM on Private and JDSearch). Both models take $\boldsymbol{u}, \boldsymbol{v}, \boldsymbol{q}$ as inputs and are trained on the same behavioral signals. They are likely to present similar results despite architectural differences. This observation suggests that relying solely on behavioral signals is insufficient for models to learn the single intended effect, as the two effects are intertwined. This highlights the necessity for disentangled effects.

When comparing 'CLK' with the 'Base' of joint modeling frameworks, it's not always advantageous to learn relevance from behavior data, as suggested by [34]. While the original literature indicates that relevance models can benefit from click data, this may not hold in the context of behavior modeling. This creates a modeling loop: 'behavior -> relevance -> behavior', which suffers from a lack of human-labeled relevance data to refine the relevance and prevent overfitting on 'behavior'. This situation calls for a design that does not depend on human-labeled data.

For CVR solutions, NISE, and DCMT, although they partly address the disentangled effect and the misaligned modeling space, they still fail to significantly enhance the joint modeling framework ('Base'). Because the click signal is observed in CVR problems, while both relevance and preference signals are unavailable in our scenario. This necessitates a modeling space reconstruction solution specifically tailored to our joint behavior modeling challenges.

The existing adaptive fusion method, PRINT, fails to consistently improve joint modeling performance. It divides the relevance effect into three segments: positive, zero, and negative effects, corresponding to Area#4&5, Area#0&3, and Area#1&2, respectively. PRINT struggles to discern the most subtle patterns within combined areas (e.g., Area#1 in Area#1&2), leading to suboptimal outcomes, which are also affected by the unstable Gumbel-Softmax used for area prediction. This calls for a more fine-grained adaptive fusion framework that provides stable predictions across areas.

The proposed DRP outperforms others by incorporating preference editing and adaptive fusion, effectively addressing the issue of entangled effects and reconstructing a fine-grained modeling space. Our work delves into the nuances of the joint modeling problem and avoids the need for human-labeled relevance data.

## 4.3 Ablation Study

We investigate core components of DRP in this section, focusing on four designed variants to study the effects of orthogonal preference editing and dual-level adaptive scoring. This variant drops the orthogonal constraint on $\boldsymbol{O}$. Specifically, it replaces Equation (6) with $\boldsymbol{e}_{p_c} = \boldsymbol{W}_2^T(\boldsymbol{W}_1\boldsymbol{e}_p - \boldsymbol{W}_1\boldsymbol{e}_r)$, where $\boldsymbol{W}_1$ and $\boldsymbol{W}_2$ are learnable

**Table 3: Overall performance comparison.**

| PM | RM | JM | Private | | | | KuaiSAR | | | | JDSearch | | | |
|---|---|---|---|---|---|---|---|---|---|---|---|---|---|---|
| | | | AUC | LogLoss | NDCG | HR | AUC | LogLoss | NDCG | HR | AUC | LogLoss | NDCG | HR |
| - | DSSM | | 0.6468 | 0.05674 | 0.1628 | 0.2867 | 0.5687 | 0.3662 | 0.4088 | 0.7085 | 0.6554 | 0.09837 | 0.1916 | 0.3356 |
| | QEM | - | 0.6491 | 0.05606 | 0.1635 | 0.2881 | 0.5669 | 0.3640 | 0.4087 | 0.7102 | 0.6564 | 0.09712 | 0.1931 | 0.3355 |
| | HEM | | 0.6684 | 0.05577 | 0.1634 | 0.2883 | 0.6146 | 0.3561 | 0.4091 | 0.7100 | 0.6560 | 0.09705 | 0.1984 | 0.3426 |
| MLP | - | - | 0.6686 | 0.05577 | 0.1632 | 0.2879 | 0.6123 | 0.3566 | 0.4098 | 0.7107 | 0.6557 | 0.09705 | 0.1961 | 0.3394 |
| | DSSM | Base | 0.6690 | 0.05572 | 0.1649 | 0.2904 | 0.6206 | 0.3556 | 0.4112 | 0.7117 | 0.6669 | **0.09670** | 0.1968 | 0.3445 |
| | | CLK | 0.6692 | 0.05571 | 0.1648 | 0.2903 | 0.6205 | 0.3556 | 0.4110 | 0.7115 | 0.6657 | 0.09682 | 0.1960 | 0.3427 |
| | | NISE | 0.6689 | 0.05571 | 0.1648 | 0.2906 | 0.6207 | 0.3555 | 0.4113 | 0.7119 | 0.6669 | **0.09670** | 0.1969 | 0.3444 |
| | | PRINT | 0.6696 | 0.05564 | 0.1644 | 0.2897 | 0.6195 | 0.3552 | 0.4084 | 0.7100 | 0.6696 | 0.09724 | 0.2018 | 0.3477 |
| | | DRP | **0.6710*** | **0.05559*** | **0.1660*** | **0.2920*** | **0.6229*** | 0.3549 | **0.4168*** | **0.7145*** | **0.6824*** | 0.09685 | **0.2052*** | **0.3620*** |
| | QEM | Base | 0.6696 | 0.05566 | 0.1642 | 0.2897 | 0.6206 | 0.3554 | 0.4112 | 0.7118 | 0.6783 | 0.09874 | 0.1979 | 0.3550 |
| | | CLK | 0.6701 | 0.05565 | 0.1662 | 0.2922 | 0.6205 | 0.3553 | 0.4113 | 0.7112 | 0.6787 | 0.09878 | 0.1960 | 0.3524 |
| | | NISE | 0.6698 | 0.05570 | 0.1657 | 0.2913 | 0.6207 | 0.3553 | 0.4115 | 0.7122 | 0.6793 | 0.09882 | 0.1963 | 0.3539 |
| | | DCMT | 0.6699 | 0.05571 | 0.1656 | 0.2913 | 0.6214 | **0.3550** | 0.4110 | 0.7122 | 0.6796 | 0.09888 | 0.1972 | 0.3537 |
| | | PRINT | 0.6689 | 0.05568 | 0.1638 | 0.2889 | 0.6208 | 0.3556 | 0.4095 | 0.7110 | 0.6707 | 0.09930 | 0.1987 | 0.3553 |
| | | DRP | **0.6707*** | **0.05561*** | **0.1663** | **0.2923** | **0.6227*** | 0.3551 | **0.4143*** | **0.7132*** | **0.6904*** | **0.09862*** | **0.2053*** | **0.3635*** |
| | HEM | Base | 0.6695 | 0.05570 | 0.1649 | 0.2903 | 0.6218 | 0.3553 | 0.4118 | 0.7117 | 0.6774 | 0.09884 | 0.2032 | 0.3616 |
| | | CLK | 0.6694 | 0.05568 | 0.1652 | 0.2905 | 0.6217 | 0.3553 | 0.4115 | 0.7114 | 0.6770 | 0.09863 | 0.1989 | 0.3546 |
| | | NISE | 0.6695 | 0.05568 | 0.1647 | 0.2901 | 0.6218 | 0.3553 | 0.4116 | 0.7117 | 0.6776 | 0.09886 | 0.2029 | 0.3609 |
| | | DCMT | 0.6699 | 0.05570 | 0.1659 | 0.2912 | 0.6214 | 0.3552 | 0.4101 | 0.7116 | 0.6778 | 0.09895 | 0.1998 | 0.3574 |
| | | PRINT | 0.6693 | 0.05568 | 0.1645 | 0.2900 | 0.6209 | 0.3554 | 0.4094 | 0.7109 | 0.6758 | 0.09902 | 0.1956 | 0.3520 |
| | | DRP | **0.6702** | **0.05563*** | **0.1663** | **0.2924*** | **0.6230*** | 0.3551 | **0.4138*** | 0.7122 | **0.6876*** | **0.09814*** | **0.2085*** | **0.3690*** |
| DCN | - | - | 0.6603 | 0.05598 | 0.1607 | 0.2839 | 0.6183 | 0.3564 | 0.4144 | 0.7126 | 0.6738 | 0.09806 | 0.2096 | 0.3634 |
| | DSSM | Base | 0.6608 | 0.05595 | 0.1609 | 0.2845 | 0.6184 | 0.3561 | 0.4132 | 0.7120 | 0.6702 | 0.09660 | 0.2022 | 0.3504 |
| | | CLK | 0.6615 | 0.05595 | 0.1615 | 0.2852 | 0.6183 | 0.3561 | 0.4131 | 0.7117 | 0.6686 | 0.09674 | 0.2009 | 0.3481 |
| | | NISE | 0.6612 | 0.05590 | 0.1613 | 0.2852 | 0.6184 | 0.3561 | 0.4133 | 0.7124 | 0.6702 | 0.09660 | 0.2022 | 0.3503 |
| | | PRINT | 0.6599 | 0.05605 | 0.1598 | 0.2830 | 0.6165 | 0.3561 | 0.4108 | 0.7103 | 0.6724 | 0.09712 | 0.2108 | 0.3629 |
| | | DRP | **0.6666*** | **0.05577*** | **0.1643*** | **0.2891*** | **0.6200*** | **0.3557*** | **0.4145*** | **0.7133** | **0.6915*** | 0.09654 | 0.2125 | **0.3702*** |
| | QEM | Base | 0.6606 | 0.05604 | 0.1610 | 0.2845 | 0.6192 | 0.3555 | 0.4118 | 0.7107 | 0.6874 | 0.09894 | 0.2143 | 0.3745 |
| | | CLK | 0.6604 | 0.05611 | 0.1612 | 0.2843 | 0.6190 | 0.3555 | 0.4118 | 0.7097 | 0.6857 | 0.09931 | 0.2120 | 0.3706 |
| | | NISE | 0.6603 | 0.05611 | 0.1611 | 0.2843 | 0.6193 | 0.3555 | 0.4119 | 0.7106 | 0.6876 | 0.09887 | 0.2142 | 0.3749 |
| | | DCMT | 0.6602 | 0.05601 | 0.1604 | 0.2836 | 0.6208 | 0.3553 | 0.4134 | 0.7120 | 0.6843 | 0.09907 | **0.2166** | **0.3776** |
| | | PRINT | 0.6606 | 0.05592 | 0.1604 | 0.2838 | 0.6202 | 0.3555 | 0.4138 | 0.7125 | 0.6842 | **0.09877** | 0.2136 | 0.3736 |
| | | DRP | **0.6663*** | **0.05566*** | **0.1632*** | **0.2873*** | **0.6243*** | **0.3547*** | **0.4163*** | **0.7142*** | **0.6942*** | 0.09882 | 0.2154 | 0.3767 |
| | HEM | Base | 0.6618 | 0.05590 | 0.1607 | 0.2836 | 0.6229 | 0.3548 | 0.4132 | 0.7133 | 0.6852 | 0.09900 | 0.2184 | 0.3798 |
| | | CLK | 0.6610 | 0.05595 | 0.1602 | 0.2830 | 0.6231 | 0.3548 | 0.4131 | 0.7124 | 0.6868 | 0.09912 | 0.2167 | 0.3766 |
| | | NISE | 0.6611 | 0.05595 | 0.1602 | 0.2834 | 0.6229 | 0.3548 | 0.4131 | 0.7129 | 0.6855 | 0.09898 | 0.2178 | 0.3792 |
| | | DCMT | 0.6603 | 0.05608 | 0.1612 | 0.2841 | 0.6216 | 0.3550 | 0.4132 | 0.7122 | 0.6874 | 0.09890 | 0.2184 | 0.3804 |
| | | PRINT | 0.6603 | 0.05599 | 0.1601 | 0.2841 | 0.6209 | 0.3551 | 0.4119 | 0.7112 | 0.6841 | 0.09875 | 0.2142 | 0.3752 |
| | | DRP | **0.6678*** | **0.05562*** | **0.1647*** | **0.2896*** | **0.6236** | 0.3545 | **0.4152*** | **0.7144*** | **0.6938*** | 0.09871 | 0.2194 | **0.3855*** |

"*" indicates the statistically significant improvements. (i.e., one-sided t-test with $p < 0.05$) over the runner-up method.

matrices with the same dimensions as $O$. **DRP-2**: This variant omits the entire adaptive fusion component to isolate the effectiveness of the remaining component, namely, preference editing. **DRP-3**: This variant excludes global adaptive fusion to assess its impact. In other words, $\hat{y}_g$ in Equation (10) is replaced by $\hat{y}$ from Equation (7). **DRP-4**: This variant skips Equation (10) to evaluate its efficacy, focusing on the effectiveness of local adaptive scoring.

We evaluate these variants on JDSearch with MLP. The result is listed in Table 4, where AUC and NDCG are presented. From this table, we can get the following conclusions. DRP-1 fails to yield valid results (omitted in the table). Without the orthogonal constraint, the editing space learned by $W_1$ may not align with the causal intervention space of relevance on preference, potentially

subtracting predictive information about user behaviors. Moreover, the learned reverse transformation $W_2$ is not necessarily the strict inverse of $W_1$, which could result in missing the original prediction space captured by $W_p$ in Equation (7). DRP-2 still outperforms the 'Base', suggesting that even with just preference editing, joint modeling can benefit from disentangled relevance and preference effects. DRP-3 and DRP-4 perform better than DRP-2, underscoring the effectiveness of global and local adaptive fusion, respectively. The original design achieves the best results, outperforming all variants. This suggests that the combination of adaptive fusion with disentangled effects greatly enhances effectiveness, highlighting the success of the reconstructed modeling space by DRP.

Table 4: Ablation study on JDSearch with MLP.

| | DSSM | | QEM | | HEM | |
| | AUC | NDCG | AUC | NDCG | AUC | NDCG |
| --- | --- | --- | --- | --- | --- | --- |
| Base | 0.6669 | 0.1968 | 0.6783 | 0.1979 | 0.6774 | 0.2032 |
| DRP | 0.6824 | 0.2052 | 0.6904 | 0.2053 | 0.6876 | 0.2085 |
| DRP-2 | 0.6671 | 0.1969 | 0.6808 | 0.2021 | 0.6796 | 0.1990 |
| DRP-3 | 0.6673 | 0.1984 | 0.6821 | 0.2040 | 0.6818 | 0.2050 |
| DRP-4 | 0.6683 | 0.1979 | 0.6832 | 0.1975 | 0.6842 | 0.2070 |

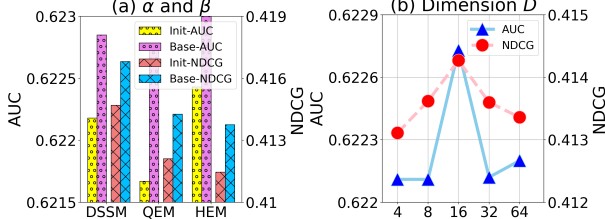

Figure 3: The hyper-parameter experiments on KuaiSAR.

### 4.4 Parameter Analysis

This section explores the sensitivity of DRP to key parameters, specifically the dimension $D$ of the low-rank projection space and the initial values of $\alpha$ and $\beta$ for the global adaptive fusion. We varied $D$ across the set $[4, 8, 16, 32, 64]$ and tested initial values of $\alpha = \beta = (1, 0.5)$, denoted as *Base* in the Figure 3 (a), and $\alpha = \beta = (0.5, 0.5)$, denoted as *Init*. We change one parameter at a time and evaluate DRP's performance on KuaiSAR using MLP.

From **Figure 3 (a)**, the distinct initialization of $\alpha$ and $\beta$ as $(1, 0.5)$ is beneficial. This initialization appears to provide DRP with the prior knowledge that positive relevance or preference usually leads to positive behaviors. From **Figure 3 (b)**, the performance shows an inverse V-shaped pattern as $D$ increases. A lower-dimensional space may be insufficient to capture the influence of relevance on preference, explaining the initial performance boost as $D$ grows. However, an overparameterized space might introduce irrelevant information. causing the performance drop.

### 4.5 Model Visualization

This section visualizes the effects of dual-level adaptive fusion described in Section 3.3. We load the checkpoint of DRP with MLP and HEM on KuaiSAR, subsequently calculating the average behavior predictions for each designated area (Area#0-5), illustrated via a heatmap in Figure 4. We classify samples with the top 20% relevance scores as relevant and the same for preference to construct areas. For illustrative clarity, we incorporate an additional Area#0 following Area#6. The modeled areas depicted in Figure 4 (I-III) correspond directly to the configuration shown in Figure 2(c). From this figure, several observations emerge: (i) The fixed fusion in Equation 7 fails to differentiate Area#2&4 and Area#1&5, collapsed as Area#3. (ii) Through global adaptive fusion, DRP successfully discriminates among four distinct areas: Area#0, Area#1&2, Area#3, and Area#4&5 (noted by similar color in the figure). (iii) The implementation of local adaptive fusion ultimately enables discerning fine-grained patterns across various areas. These findings are in accord with our theoretical analysis in Section 3.3, underscoring the rationale behind our design.

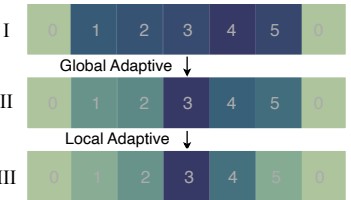

Figure 4: Model visualization of adaptive fusion.

## 5 RELATED WORKS

We categorize the related work into two main areas: e-commerce search models and solutions to collapsed modeling spaces.

E-commerce search models, also known as product search models, can be divided into three categories: ad-hoc search models, personalized product search, and the development of relevance models. Ad-hoc models [3, 18, 26] conceptualize e-commerce search as a user-independent relevance modeling problem. While personalized search endeavors to synthesize user behavior sequences to generate personalized results, which is enhanced through the utilization of knowledge graphs [2] and transformers [1, 5]. Xiao et al. [33] was the first to articulate the notion of relevance modeling, which has inspired numerous subsequent efforts aimed at designing pre-trained relevance models to be integrated within a preference modeling framework, operating as a joint behavior modeling approach. Techniques such as knowledge distillation [20], self-supervised learning [8], and pretraining pipelines [35] are employed in this context. Nonetheless, these frameworks currently exhibit a heavy reliance on laboriously collected relevance data; although some enhancement methods based on click data have been proposed [34], there remains unexplored for the joint modeling framework to address latent issues or augment its applicability. A relevant contribution might be PRINT [16], which personally merges the relevance effect during the fusion phase in a vague manner. Our study examines the challenges posed by the collapsed modeling space problem within the joint framework, and we introduce DRP, which employs fine-grained, explicit adaptive fusion for easy deployment across diverse systems without relevance data.

Another relevant research area is the solution for the collapsed modeling space. Despite substantial attention from CVR prediction studies [17, 29, 37] and approaches addressing impression bias [21, 22], joint modeling strategies remain limited explorations. We are pioneering a thorough analysis of the collapsed modeling space issue within the context of relevance-preference joint modeling.

## 6 CONCLUSION

This paper introduced DRP, a novel joint modeling framework for e-commerce search systems. We begin with a comprehensive review of the existing frameworks and point out the collapsed modeling space problem. DRP innovatively edits the preference representation to exclude the relevance influence on it, thus reconstructing the desired behavior modeling space with only direct effects from the preference and relevance. It then captures nuanced patterns within the fine-grained modeling space by dual-level adaptive fusion strategy. With these components, DRP could be applied with various preference and relevance backbones without laborious collected relevance data. Our extensive experiments have supported the superiority of DRP in search scenarios.

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
