# OpenReview forum: "Behavior Modeling Space Reconstruction for E-Commerce Search"
_ACM.org/TheWebConf/2025/Conference — WWW 2025 Oral_

### Official Review · Reviewer_4duU · 2024-11-17

**Novelty:** 5
**Technical Quality:** 4

**Review:**

This paper introduces the DRP framework to address the collapsed modeling space issue in e-commerce search, caused by the entanglement of user preferences and query-item relevance. it uses preference editing to disentangle these effects and dual-level adaptive fusion to model nuanced user behavior patterns. Experimental results on public and proprietary datasets demonstrate its effectiveness without relying on human-labeled data.

Pros:
- This paper provides a clear analysis of existing methods using causal graphs and Venn diagrams and review two important issues: entangled preference and relevance effects and collapsed modeling space.
- The proposed framework DRP, which includes preference editing and adaptive fusion, is overall reasonable for solving the three problems mentioned in Section 2.2.
- Experiments demonstrate the effectiveness of DRP from diverse aspects, including a comparison between baselines, ablation studies, and parameter analysis.
- Link to the code is provided for reproduction.

Cons:
- In Figure 1(a), there is a red path from R (relevance) to P (preference). Can there be a path from P to R? The paper didn't mention it.
- Since a provite dataset is used to evaluate the performance of DRP, the paper didn't mention the possibility of DRP being deployed in real-world E-Commerce systems.
- In real-world applications, efficiency is an important issue. The paper should talk about the efficiency between DRP and other baselines.
- Only MLP and DCN are chosen as the PM model, which I think is limited. I finf that the code also includes DeepFM and DESTINE, which is not included in the paper.

**Questions:**

See the Cons. Other questions related to the experiments:
- How does the general hyper-parameters, such as learning rate, embedding sizes, and special hyper-parameters in other baselines (if exists) are tuned?
- The title is "Behavior Modeling Space Reconstruction for **E-Commerce Search**". The performance of DRP was verified on only three datasets, among which KuaiSAR is not related to e-commerce. Have you considered including more e-commerce search datasets?
- In Table 2, the number of users in JDSearch is missing. Is this dataset missing user information? How do you model user behaviors?

**Reviewer Confidence:**

3: The reviewer is confident but not certain that the evaluation is correct

**Scope:**

4: The work is relevant to the Web and to the track, and is of broad interest to the community

---

### Official Review · Reviewer_sLSN · 2024-11-22

**Novelty:** 5
**Technical Quality:** 5

**Review:**

𝐒𝐮𝐦𝐦𝐚𝐫𝐲:
The paper introduces DRP (Disentangled Relevance and Preference), a framework aimed at enhancing e-commerce search systems by addressing challenges in behavior modeling. Traditional approaches often entangle user preferences and query-item relevance, resulting in a collapsed modeling space. The proposed DRP disentangles these effects using Preference Editing and improves scoring precision through Adaptive Fusion. Extensive experiments on public and proprietary datasets demonstrate the superiority of DRP in predictive accuracy and ranking tasks without relying on human-labeled relevance data.

𝐒𝐭𝐫𝐞𝐧𝐠𝐭𝐡𝐬:
- 𝐒𝟏: The paper identifies a critical issue in collapsed modeling spaces for e-commerce search. By combining causal graphs and Venn diagrams, it provides a clear and compelling explanation of the problem.

- 𝐒𝟐: The proposed Preference Editing effectively disentangles relevance from preference, and the Dual-Level Adaptive Fusion offers nuanced, fine-grained behavior modeling. These innovations directly address the identified challenges.

- 𝐒𝟑: The authors validate their framework on three datasets (two public and one proprietary), demonstrating versatility across different contexts. The significant performance improvements, coupled with rigorous ablation studies, underline the robustness of DRP.

- 𝐒𝟒: The framework's independence from human-labeled data enhances its applicability to diverse e-commerce platforms, especially those lacking extensive resources for data labeling.

𝐖𝐞𝐚𝐤𝐧𝐞𝐬𝐬𝐞𝐬:
- 𝐖𝟏: The descriptions of the Preference Editing and Adaptive Fusion processes are mathematically dense, making it challenging for readers outside the immediate research domain to fully grasp the mechanisms.

- 𝐖𝟐: While the evaluation compares DRP against several baselines, the selection of these baselines appears narrow. Including a broader spectrum of recent approaches, especially those emphasizing disentangled representations, would strengthen the claims.

- 𝐖𝟑: The paper mentions computational efficiency but does not explicitly quantify DRP’s scalability for extremely large datasets or real-time applications. Providing computational complexity analysis or latency metrics would make the results more actionable.

- 𝐖𝟒: Ethical Implications: Although the method is robust, its potential impact on reinforcing biases in recommendation systems or influencing consumer behavior disproportionately is not addressed.

𝐒𝐮𝐠𝐠𝐞𝐬𝐭𝐢𝐨𝐧𝐬 𝐟𝐨𝐫 𝐈𝐦𝐩𝐫𝐨𝐯𝐞𝐦𝐞𝐧𝐭
- Simplify or supplement the mathematical formulations with intuitive examples or analogies. This would help bridge the gap for readers unfamiliar with advanced causal modeling techniques.

- Include additional state-of-the-art models, particularly those addressing similar challenges in disentangled behavior modeling.

- Report computational costs, training times, and inference latencies to evaluate DRP's feasibility in large-scale, real-time environments.

- Discuss potential biases in reconstructed modeling spaces and provide strategies to mitigate adverse impacts.

**Questions:**

- How does DRP handle edge cases where the preference or relevance signals are highly ambiguous or contradictory?

- Could the preference editing technique be extended to other domains, such as personalized recommendations or voice-based searches?

- How does the framework handle dynamic user behavior patterns, such as rapidly changing preferences in seasonal e-commerce trends?

- What measures have been implemented to ensure the fairness and neutrality of the disentangled predictions?

**Reviewer Confidence:**

3: The reviewer is confident but not certain that the evaluation is correct

**Scope:**

4: The work is relevant to the Web and to the track, and is of broad interest to the community

---

### Official Review · Reviewer_V3E9 · 2024-12-01

**Novelty:** 5
**Technical Quality:** 5

**Review:**

This paper focuses on E-Commerce Search. Specifically, it introduces DRP, which edits the preference representation to exclude the relevance influence on it, reconstructing the desired behavior modeling space with only direct effects from the preference and relevance. Experiments on three datasets demonstrate the effectiveness of the proposed method.

Advantages:
1. This paper provides a theatrical analysis to show the necessity of addressing the collapsed modeling space with disentangled relevance and preference effects.
2. Experiments on three datasets demonstrate the effectiveness of the proposed method.
3. The code has been open-souced.

Limitations
More justification is needed. For example, what are the differences between this work and other studies into neural ranking and learning-to-rank studies? Are methods about neural ranking methods (e.g., DPR, ANCE, Splade) and learning-to-rank methods applicable to the e-commerce search scenario? If so, why not use them as baselines?

**Questions:**

What are the differences between this work and other studies into neural ranking and learning-to-rank studies? Are methods about neural ranking methods (e.g., DPR, ANCE, Splade) and learning-to-rank methods applicable to the e-commerce search scenario? If so, why not use them as baselines?

**Reviewer Confidence:**

3: The reviewer is confident but not certain that the evaluation is correct

**Scope:**

4: The work is relevant to the Web and to the track, and is of broad interest to the community

---

### Official Review · Reviewer_BhQK · 2024-12-02

**Novelty:** 5
**Technical Quality:** 5

**Review:**

This paper submission a unified framework for behavior modeling in e-commerce search, addressing limitations in joint modeling approaches. It identifies three critical issues: (1) collapsed modeling spaces caused by ignoring interactions between relevance and preference, (2) static integration of relevance and preference effects, and (3) biased optimization resulting from sequential training processes.
To resolve these challenges, the authors propose the Disentangled Relevance-Preference (DRP) model, which employs calibrated preference prediction to disentangle relevance effects from preference. Experiments on two public datasets and one private dataset demonstrate the superior performance of DRP.
The paper is well-structured, though certain sections are challenging to follow. Its originality lies in disentangling the effects of relevance from user preferences. Extensive experiments conducted on three datasets demonstrate that the proposed approach is effective when compared to several strong baselines.
My main issues (questions)  with the paper are the following:
- The authors describe assumptions about collapsed modeling spaces with six areas but (I do no see) do not clearly demonstrate how these assumptions translate into experimental outcomes. Its practical implications in the experiments are not evident (at least not clear for me).
- There is insufficient detail regarding the features used to construct the user (u) and item (v) representations. The paper mentions that item features—"like titles or product names—are converted into dense representations using a text encoder, combined with other item features to form embeddings. It also references queries involving term frequency”. However, it is unclear which specific features were utilized and whether the baseline models used comparable features.

**Questions:**

My main questions:
- The authors describe assumptions about collapsed modeling spaces with six areas but (I do no see) do not clearly demonstrate how these assumptions translate into experimental outcomes. Its practical implications in the experiments are not evident (at least not clear for me).

- There is insufficient detail regarding the features used to construct the user (u) and item (v) representations. The paper mentions that item features—"like titles or product names—are converted into dense representations using a text encoder, combined with other item features to form embeddings. It also references queries involving term frequency”. However, it is unclear which specific features were utilized and whether the baseline models used comparable features.

**Reviewer Confidence:**

1: The reviewer's evaluation is an educated guess

**Scope:**

4: The work is relevant to the Web and to the track, and is of broad interest to the community

---

### Official Review · Reviewer_yEft · 2024-12-02

**Novelty:** 4
**Technical Quality:** 4

**Review:**

This research tackles issues in modeling search behavior in e-commerce by presenting the DRP framework, which disentangles relevance and preference effects via preference editing and adaptive fusion. Through the reconstruction of the modeling space, DRP seeks to improve search accuracy independent of human-labeled relevance data. Experiments using public and proprietary datasets exhibit substantial enhancements in performance compared to previous methodologies.

Strengths:
1. The method interprets the existing problems in joint relevance and preference model training using causal graphs and Venn diagrams and proposes disentangled joint training with adaptive fusion.
2. Extensive experiments on multiple public datasets and a proprietary dataset demonstrate strong performance improvements over existing approaches.
3. The paper is well written.

Weaknesses:
1. The contribution section describes the work as a theoretical framework based on causal graphs and Venn diagrams, but the analysis comes across as more heuristic and practical in nature. Adding more formal details or adjusting the wording to better reflect its heuristic approach could make the claim more accurate and convincing.
2. The paper mentions that training the model for relevance first and then  preference might introduce a bias toward relevance. While this might be a reasonable assumption, not always hold true, as the outcome depends on the specific training strategy of preference. Providing proof to back up this assertion, like a tiny experiment, would strengthen the case and validate its veracity.
3. The proposed approaches like preference editing and dual-level adaptive fusion may enhance computational complexity. Smaller e-commerce platforms with little resources may struggle. A description of computational efficiency, including training time and resource needs, would make the work more useful.

Questions:
1. In section 2.1 “ The joint model first obtains the query and item representations, q and r, using the text …. ” is this r or v?
2. How can it be ensured during training that the O matrix maintains its orthogonality?
3. In section 4.3 “ In other words, yˆg in Equation (10) is replaced by yˆ from Equation (7)” is it Equation (8) instead of (7)?

**Questions:**

As mentioned above in Weaknesses and Questions.

**Reviewer Confidence:**

2: The reviewer is willing to defend the evaluation, but it is likely that the reviewer did not understand parts of the paper

**Scope:**

3: The work is somewhat relevant to the Web and to the track, and is of narrow interest to a sub-community